# Towards Professionalism and Police Legitimacy? An Examination of the Education and Training Reforms of the Police in the Republic of Ireland

**Jeremy Thompson [1,* ] and Brian Payne [2]**

1   School of Education, Ulster University, Cromore Road, Coleraine BT52 1SA, Northern Ireland, UK
2   School of Applied Social & Policy Sciences, Ulster University, Shore Road, Newtownabbey BT37 0QB, Northern Ireland, UK; b.payne@ulster.ac.uk
*   Correspondence: jeremythompson1@live.co.uk

**Abstract:** In this paper, we present a thought piece examining recent core policing reforms introduced in the Republic of Ireland (ROI), responding to a perceived crisis of legitimacy, for An Garda Síochána (AGS) (translated: 'The Guardian of the Peace'). Central to this process is the critical reform of the education and training of police and their relationship to the professionalisation and legitimacy of policing. In this paper, we put forward an explorative analysis of the potential link between the professional education of police and their perceived legitimacy. A literature review was carried out on the reform process, including the related elements of police education, training, professionalisation, community policing, police legitimacy, code of ethics (CoE) and police culture. We consider the espoused ambition to professionalise policing via processes including the provision of professional learning in universities and how this might be deemed to contribute (or not) to legitimacy. While no empirical research to date has been carried out on these specific reforms in the ROI, the reform recommendations had several resonances with broader examination of the themes and challenges (in particular police ethics and culture) associated with reform of democratic policing in other jurisdictions, particularly with respect to increasing professional learning and perceived police legitimacy.

**Keywords:** education; training; legitimacy; profession; professionalisation; police; policing; reform; community

## 1. Introduction

### 1.1. The Police Professional and the Place Model

Exploring what it means to be a professional is a task in itself. Bourdieu suggests the following:

'Profession' is a folk concept which has been uncritically smuggled into scientific language and which imports into it a whole social unconscious. It is the social product of a historical work of construction of a group and of a representation of groups that has surreptitiously slipped into the science of this group. [1] (pp. 242–243)

Similarly, Clarke [2] focuses on both the currency and slipperiness of 'professionalism', citing Bourdieu's assertion of a profession as 'dangerous' yet resting in its 'appearance of neutrality' [1] (p. 242). Clarke [2,3] proceeds to define professionals as those with trustworthy expertise built up by a process of career-long professional learning. Her place model seeks to examine the relationship between this professional learning (building expertise and trustworthiness—on a notional

horizontal axis) and public esteem (on a notional vertical axis), uncovering dystopias (including unprofessionalism and professionalisation) in many aspects of the relationship but also highlighting an ideal of learned (and learning) and trustworthy inclusive professionalism, which is highly esteemed by the public [2,3].

Such factors resonate strongly with critical aspects of the policing literature; therefore, it is unsurprising that previous research has exercised Bourdieu's key concepts of field, capital, habitus and symbolic violence in the context of university police education, which in turn is perceived to enhance the status of police learning and professionalism [4]. However, as described below, mapping police education is not without problems [4] (p. 1), with its impact generally unexplored and scant of empirical examination [5] (pp. 1–2, 12). In this paper, we hope to provoke further debate by applying Clarke's place model to the concept of police education and professionalism. In this conception, police legitimacy replaces esteem on Clarke's vertical axes, and professional learning is the horizontal axis. While the relationship between the two can never take the form of a panacea, the potential discrepancies and concurrencies are worth exploring in this early think piece.

### 1.2. Police Professionalism, Legitimacy and Community Policing

While acknowledging Marenin's consideration that the necessities for democratic policing fall into three principle conditions, that of accountability, professionalism and legitimacy [6] (pp. 109–110), in this paper, we will focus mainly on the latter two, as we argue that they resonate most with the idea of police learning. To maintain and develop police legitimacy, certain fundamental factors such as prejudicial neutrality, moral consensus, responsiveness to society and the representation of officers to the society they serve, are critical [6–8]. The broader notions of professionalism and professionalisation have evolved together with organisations and the educational learning processes that prepare people to work in them [9] (p. 154), which, for policing, relates to a range of skills, values and standards of fairness, integrity and human rights [6] (p. 109). In the last century, the police professional has experienced a demanding change moving from reacting to crimes to encompassing the broader gambit of 'preventing' crimes and problem-solving [7], detailing deeper cooperation between the police and their communities and emphasising a shift in the coproduction of public safety towards a model of community-orientated policing [7,10–12]. In America, Williams et al. [9] (pp. 9, 156–157), having observed transitions in paradigms of police professionalism, note that it is the support for community policing that shows the most growth. In Northern Ireland, the Independent Commission on Policing for Northern Ireland (ICPNI) [13], referred to as 'The Patten Report', regarded internationally as a template for democratic policing [14,15], arguably elevated community policing so it is not solely the responsibility of the police but rather involves the active support, involvement and consent of the communities based on the principle of 'policing with the community' (PwC) [13] (pp. 6, 7, 40, 82)—an ambition that echoed Flanagan's view that community policing 'is much too important, and too impactive on all our lives to be left to the police alone' [16] (p. 2).

Reforms in the professionalisation of policing and the embracing of a community policing model, including the important related factors of education and training, have not been advanced in isolation and remain the subject of debate in the democratic world. Proponents reason that better-educated officers will exhibit less bias and provide more objective and reflective decision making, linked to enhanced communication and problem-solving skills central to the concept of community policing [17]. However, the benefits to both future student officers and recruits studying at universities remains underexplored [5] (p. 2).

### 1.3. Policing in Ireland and Recent Reform

Ireland as a country was established in 1922 as the Irish Free State within the British Commonwealth, becoming a sovereign state with its own constitution in 1937 and severing its last official link with the United Kingdom in 1949. It has remained an economically productive member of the European Union since 1973, with an economy mainly resting on technology and agriculture [18] (p. 350). It has a

population of 4.7 million (2016 census) with non-Irish nationals representing 0.5 million. Residents who do not speak English as their first language number 0.6 million, while 78.3% of the population identifies as Catholic. The average age in Ireland is 37.4 [19], and as of February 2019, there were 13,977 sworn An Garda Síochána officers serving the public [20].

The origins of An Garda Síochána (AGS), although relatively recent, are similar in many ways to the experiences of other jurisdictions and share similar problems. Described previously as colonial, AGS was devised to protect the English establishment and subdue the unruly masses [18]. While the foundations of modern-day policing can be said to be found in the reforms of Sir Robert Peel, it is perhaps important to note that this Peelian model arguably extends from Peel's experience, gained with the Irish Peace Preservation Force in maintaining control of colonial Ireland prior to 1922 [18,21].

Ireland has been identified as the first new democracy of the twentieth century, and the central role that AGS played in the emergence of the state has left it closely associated and loyal to the state institutions, which it has felt mandated to protect. As Manning states, 'the police were and are an accepted arm of government' [18] (p. 353) and can be viewed as a shallow reflection of Anglo-American police forces, being 'bottom-heavy', with officers being moderately trained in a formal setting and inadequately supervised when dealing with complex policing matters. Approaching its centenary, AGS' history has been punctuated by a range of miscarriages of justice and scandals, which have tested the perceived legitimacy of the organisation but have brought about little change, with culpability being placed more so on the activities of wayward officers rather than systemic deficiencies requiring reform [18,22–24].

However, in 2002, in what is described as a watershed in Irish policing history, the government established the Morris Tribunal in the wake of serious allegations of policing misconduct. Its findings after 686 days of hearings and 4000 pages of script highlighted widespread corruption and malpractice within AGS, concluding that significant reform was required if policing legitimacy was to be restored [21,23,25]. The tribunal heralded a raft of changes, some institutional, others driven mainly by the 2005 Garda Síochána Act, the purpose collectively being to address the growing deficit in police legitimacy [24,25]. Amongst the more notable reforms was the establishment of a Professional Standards Unit within AGS and the overhaul of both the promotion and recruitment processes: in the former, to fairly promote those with the competencies required of the role and, in the latter, to encourage a more diverse and representative police service. Neglected engagement of communities was sought through newly established joint policing committees supported by a new community policing model, which was to 'renew, reinvigorate and re-structure' this new community partnership approach by harnessing and revitalising the community culture within AGS [25] (pp. 488–489).

The Tribunal also concluded from evidence that AGS was falling short of the standards of discipline required in policing and recommended the introduction of a code of ethics and the expectation that officers should exhibit in their duties the professional traits of equality, fairness, justice, respect and continuous professional development. It also identified that many of the internal Garda systems had remained unchanged for decades and established an Independent Inspectorate to report on improving the effective and efficient standards of the Gardaí as measured against international best policing practices, including the remit to report on any aspect of Garda administration or operations. Lastly, an independent Garda Síochána Ombudsman Commission (GSOC), with powers to investigate and pursue prosecutions against complaints received from the public regarding the behaviours of Garda and promote the organisational accountability of AGS, was established [25]. Ten years later, further reforms were established by the 2015 Garda Síochána Act [26], the most central being the introduction of a Policing Authority [24].

In an analysis of these institutional changes, Manning [18] (p. 357) draws a contrast between Walsh's 'quite pessimistic' [27] examination of the success of the GSOC and his own more optimistic view on the impact of the Independent Inspectorate under the guidance of Kathleen O'Toole. However, he notes that the reports from the Independent Inspectorate were merely advisory and reemphasises that AGS still remains sheltered from rigorous scrutiny due to its close connection to the formation of the

state. In summary, he concludes that approaches towards police accountability in Ireland have been ineffectual, quoting from Walsh, 'Police accountability in Ireland, in short, is neither to the people at large, nor to any collection of groups, and has never been so' [27]. Mulcahy provides additional analysis on what proved to be an inability of these reforms to stem further policing scandals [24,28], which eventually created a political crisis leading to the 'Commission on the Future of Policing in Ireland' (CFPI) [29,30]. He contends that despite the opportunity to observe other world-leading models, such as that in its bordering neighbour Northern Ireland, those comparable institutions in the ROI were 'marked by foot-dragging and a dilution of the very powers which made the Northern Irish institutions so robust and so worthy of emulation' [24]. He further argues that the necessary political will did not exist at the inception of those institutions to equip them with the powers and function to deliver on their obligations or the public's expectations.

The CFPI [30] was established in 2017 during a further controversial period in the history of AGS [31–33], with Conway suggesting that this period reflected both a crisis of legitimacy and a unique opportunity to reform [34] (pp. 1, 8). Problems became apparent throughout AGS when in 2012 'whistle-blower' Gardá officers highlighted the abuse of a road traffic offence penalty points system, their behaviour branded by the Garda Commissioner as 'quite disgusting' [35], forcing the Commissioner to resign [31,32]. The succeeding Commissioner was then embroiled in allegations that she had attempted to discredit the 'whistle-blowers', linking one of them to unfounded allegations of child sex abuse. This Commissioner subsequently retired amidst further controversy [31,36]. Damaging factors included the following:

- An admission that 937,000 non-existent alcohol breath tests were simply concocted, presenting ethical concerns around AGS [31,37,38].
- An admitted error, leading to the wrongful conviction of 15,000 motorists for minor traffic offences [31,39].
- Alleged bugging and recording of the offices of AGS Ombudsman Commission and calls between those in police custody and their legal representatives [40,41].
- Continued concern about the quality and accuracy of crime statistics [42,43].
- Serious allegations of financial misconduct at the sole Garda training college [44].

The subsequent report 'The Future of Policing in Ireland' [30] (TFPI), published in late 2018, represents the most significant core reform process in the history of AGS with the Chairperson, Kathleen O'Toole, making the following key remarks at its launch:

'We have spent over a year listening to the people of Ireland, and the police . . . the message was loud and clear. Everyone wanted more Gardá working in and with the community. They wanted a . . . professional police service...Communities around the country told us that they attach great importance to community policing...We recommend that all police service personal in the districts will be community police. This is the backbone of police work and the police mission...We have focused on transformative changes that will support those people serving communities. A more effectively managed police service will instil a culture of professionalism, beginning with recruit training and carrying through the careers of everyone in the organisation'. [45]

Mulcahy [46] (pp. 14–17) reminds us that 'reform', along with its two overlapping elements of 'representation' and the 'response' of society, is fundamental to the process of police legitimation. It would, therefore, be difficult not to contend that the suggested reform process of education and training, as it relates to the professionalisation and legitimacy of AGS, presents a current and significant opportunity to examine policing in Ireland but also to broaden understanding towards the potential application of such reform processes in other jurisdictions. However, examination is hampered due to the recent implementation of these reforms, relative to the timing of this paper, which have not been subject to any identified, published, empirical research to date. Moreover, current empirical literature

available on police culture within AGS is described as limited by Conway [34] (p. 3), while Manning notes that AGS has not been studied ethnographically by academic observers [18] (p. 455).

Nonetheless, this paper undertakes a literature review of the reform process in AGS, scrutinising the key publications and reports available, identifying and focusing on the elements and key themes that are both relevant and central to the paper. A broader review of international literature was then conducted examining the specific themes of police education and training and their relationships to core elements of reform including community policing, police legitimacy, professionalisation, a code of ethics (CoE) and police culture. At the heart of this paper is the broad examination of the potentially valuable but as yet untested relationship between police legitimacy and professional learning (in particular, where the latter is based within higher education), set against the critical context of community policing in AGS, which has yet to be developed or analysed.

## 2. Professional Learning and Legitimacy

### 2.1. Learning for Community Policing

The effectiveness of community policing both as a means for increasing trust and confidence in policing and, significantly, the perceived legitimacy of the police is now widely accepted by reformers and policy influencers alike, for example, see the UK's College of Policing's [47] citing of a systematic review from Gill et al. [48]. The popularity of community policing has, in part, been derived from research that successfully critiqued traditional models of crime fighting such as preventing and detecting crime, patrolling in cars and upholding public order [8]. However, scholars have also argued that community policing has evolved as a response to public perceptions of an aloof and legalistic police, ill-prepared to accomplish the primary mandate of preserving and protecting the citizenry [49]. Community policing may, therefore, be conceived of as 'the collective answer to abuses of power, lack of effectiveness, poor public confidence, and concerns about legitimacy' [50] (p. 47) [8,49].

Gleeson and Byrne contrast the rise of community policing in the United States of America (USA) in the last few decades, where it has become the 'dominant philosophy' in policing, with Ireland, where it was not until 2010 that AGS introduced its first community policing model, even though it could be argued that AGS had exhibited some of the fundamentals of a community policing ethos, as, from its formation in 1925, it has been an unarmed force relying on community support to carry out its functions [51] (pp. 70–71). Similarly, the TFPI report acknowledges this transformative shift towards community policing across the democratic world and identifies that neither the structure nor the practices of AGS reflect this [30] (pp. 6, 7, 17). Responding to the TFPI report and its focus on community policing, the Garda Inspectorate published its Policing with Local Communities (PwLC) report in 2018 [52], establishing critical actions to support effective policing with communities. A summary of some of those relevant key actions is presented below:

1.  Collaboration with universities to undertake academic research towards the development of evidence-based approaches to the core policing reform issues was recommended [52] (p. 7).
2.  Several community policing models were identified in operation; while principle to them were internationally recognised elements of community policing, there was lack of clarity and purpose. The PwLC recommended one clear community policing strategy with vision and purpose [52] (p. 10).
3.  Inconsistent delivery of community policing, with long-term problem-solving being poorly understood and infrequently used and with most community policing officers having not received training in problem-solving, was found. In response, it was recommended that community policing be embedded within continuous professional development (CPD) and a national training program [52] (pp. 23, 25).
4.  Development of trained and skilled community policing teams in all areas to provide long-term problem-solving was also recommended [52] (pp. 26, 62).

Mulcahy's submission to the CFPI called for community policing to be at the core of policing in Ireland, supported and reflected in training content [24] (p. 7), and while AGS' community policing strategy still remains unpublished [52] (p. 4), the PwLC examined the '*Final Report of the President Task force on 21st Century Policing*' [53], which provided descriptors of community policing, including the following:

> Police interventions which are implemented with strong policies and training in place, rooted in an understanding of procedural justice, which in turn contributes to police legitimacy. [52] (p. 62)

Such views resonate with the American context, where Paynich argued previously that the origin for better-educated police did not have its genesis within community policing or professionalisation but rested with key campaigners, such as Peel in England in the 1800s and Vollmer in America in the 1900s, with reoccurring recommendations for higher-level education appearing in subsequent reform movements. She contends that such reforms, while seeking to professionalise police, actually eroded officer discretion and increased the centralisation of policing, weakening that relationship between the police and citizens. Secondly, as the range of problems police officers negotiated broadened and became more complex, the less effective purely legal-based remedies became, creating a requirement for both community involvement and improved police–community relationships. The community policing model utilised, she argues, set about addressing such displacement, enhancing relationships with the communities and working with citizens to problem solve [17]. Paynich identifies how the reform of both police professionalisation and community policing can generate and encourage a '*need*' for higher education, as the lower ranks are given this autonomy to problem solve within communities, but then poses whether or not higher education aids the successful enactment of such reforms [17] (pp. 8, 19). Addressing her own question, she cites Paoline et al. [54] (p. 598), who observed that small but varying and decipherable degrees of attitude existed amongst officers towards community policing, with more educated officers reporting more positive views. She also cited Chappell [55], who identified that those less well-educated recruits did not perform as well as more educated officers in community policing curriculums. Radelet and Carter reinforce this requirement for higher education to effectively implement community policing, stating the following:

> Given the nature of this change, the issue of college education is even more critical. The knowledge and skills officers are being asked to exercise in community policing appears to be tailored to college preparation. [56] (p. 156)

Goldstein notes that the community policing work environment delivers to those more highly educated officers a more self-satisfying scape in which to exercise their creative abilities towards problem-solving [57]. In reforming and transitioning to community policing, Roberg and Bonn [11] (pp. 476–478) argue that the potential of higher educated officers should be employed, identifying a common thread of enhanced skills from higher-level education delivering enhanced skills, which are necessary for the effective delivery of community policing and, as will be discussed below, also bring real benefits for police legitimacy [58].

### 2.2. Police Legitimacy

Legitimacy can be expressed as the justifying of a state of affairs so that it becomes accepted socially, while being legitimate is behaving in a way that is regarded as having been founded on valid and justifiable grounds [46]. The legitimation process, therefore, presents a unique opportunity to appreciate how a set of relationships can come to dominate others [46] or, as Gramsci defines it, the particular workings of states where a specific group is successful in having other societal groups accept its values [59]. Societies depend on the police at their core to ensure that the elements of democracy are defended and that order is maintained [60,61]. Police legitimacy, therefore, according to Tyler, [62] is a property enjoyed by the police when societies view the police as just and submit to them voluntarily,

as observed in the actions and motives of individuals [63], while Manning likens it to an unwritten agreement between the police, the public and society, encompassing those implicit expectations and common responsibilities that draw together social stability [18] (pp. 349–350). For citizens, police legitimacy is therefore both the acceptance of decisions from a lawfully established authority and the enforcement of that law, if required [64,65]. However, both Tyler and McEwen et al. [62,66] observe how compliance with these decisions is either based around fear of punishment or self-interest, providing citizens with moral values in compliance. Citizens can be at odds with these decisions yet simultaneously view those decisions as being made by a legitimate authority, Reiner, summarising, states the following:

> Police legitimacy means, at a minimum, that the broad mass of the population . . . accept the authority, the lawful right, of the police to act as they do, even if disagreeing with, or regretting some specific actions. [8]

As a legal authority, the legitimacy of a police service is therefore dependent on its ability to demonstrate to the public why its exercise of powers is rightful and why the public should, in turn, choose to obey, cooperate and comply with its request [67–69]. A critical factor in achieving cooperation of this nature has been the embracing of democratic policing models, which concentrate on the notion of protecting human rights as their core responsibility [13,49]. Such approaches focus on professional and ethical standards, codes of conduct and processes to ensure the adoption of such standards into the practices and culture of police, with efforts to reform these democratic models tending to be based on the community policing model [6] (p. 110).

Marenin [6] explains, that while the process of defining what democratic policing should resemble in its culture, practices and operational policies is generally agreed upon, challenges remain in the creation and implementation of sustainable reforms, including education and training. He holds that such reforms will only become sustainable when the existing features against which they work are nullified, requiring a sense of knowing of what to dispense with and what to replace it with. He supports Karstedt [70], that successful models of reform can only serve as examples and resources to be adapted and shaped to mould to the variances of other societies, which is a template evidenced in Connolly's submission on community policing to the Irish government [71].

Bringing about reforms of this nature in the Irish context is a sizable undertaking. Manning [18] argues that AGS' close affiliations to the origins and formation of the 20th-century Irish state and central government have mandated it to protect the state's interests, rendering AGS sacred, legitimate and resistant to reform. This near semi-sacred status has also protected AGS from public opinion even in the face of scandal over its history, and he points out that they have subsequently failed to benefit from sustained and meaningful reform, despite a growing scrutiny. Bowden and Conway also point to a long history of unflagging crises that have challenged and strained police legitimacy [21,22]—a sustaining principle of which is moral consensus [8], highlighting that the laws that the police represent and enforce must relate positively to the broader held moral values of society [6], which, arguably, within Irish policing, have been imbalanced. Connolly's historical reflection on the government in Ireland was one that sought to create a policing service based on public consent but only succeeded in creating a model symbolised and characterised by the absence of consent and public accountability [72].

The principle of 'policing by consent' [73] or that public consent towards policing must exist to deliver legitimacy, acting as a keystone in the relationship between police and citizens [74], requires it to be vigorously pursued by the police rather than being an artificial or theoretical strategy [75]. Reforms in pursuit, however, can suffer from apathy as they become embedded and institutionalised within policing culture, and that initial effectiveness, while delivering on police legitimacy to citizens in the short term is incapable by itself of maintaining that legitimacy [6] (pp. 112–113). It might therefore be argued that professional learning must be ongoing and career-long to make a sustained positive impact on police culture and legitimacy.

*2.3. Education, Training and Continuous Professional Development*

While the TFPI report identifies police education and learning as critical to the transformation and development of professional policing in Ireland, it crucially observed that it had been neglected, devoid of either a specific strategy or budget, with an absence of both resulting in an embargo of police recruitment for nearly six years and a near-zero delivery of in-service training, impacting the function and delivery of professional policing [30] (p. 69). Consider the following examples:

- Approximately 700 untrained detectives were identified, some of whom had up to 10 years of experience in investigating serious crimes yet had no specialised training and lacked CPD. Despite a capacity to train 60 detectives annually, between 2010 and 2013, only 88 detectives were trained, identifying a potentially increasing backlog [43,76,77].
- Under use of fingerprinting opportunities was found; in 2012, of 26,149 people who should have had their fingerprints taken, only 8147 were fingerprinted, and 69% were not [76,77].
- The Garda Inspectorate analysing Garda foundation training between 2000 and 2009 estimated that Garda recruits only spent 25% of available training time on operational policing and crime investigation skills, which was less than the total time spent on language skills (12%), physical exercise (17%) and study assessments (9%), with no time given to the practical interview of suspects. [43,76] (pp. 25, 244, 247, 248).
- Between 2005 and 2009 and suspension of recruitment, there was a large increase in the numbers recruited (275 per quarter), accompanied by a change in the training delivery for recruits. Its focus incorporated little practical training with a move away from small classes to presentations to 190 students at one time, with minimal assessment and screening processes. The Garda Inspectorate identified that this has led to difficulties for recruits emanating from this period, recommending a specific training needs analysis for this cohort [76] (pp. 248–249).

The TFPI report identifies a degree-level foundation program introduced in AGS in 2014 and validated by Limerick University, Ireland as the current core to recruit training. It is delivered entirely within the police training college with nearly all staff being sworn police officers and initially involves 32 weeks at the Garda Training College, at which point recruits are attested. This is followed by 34 weeks of supervised on-the-job experiential learning at designated Garda stations. In the third and last phase, recruits work as probationer Garda officers with short periods spent at the training college, with the whole process lasting two years [30] (p. 70). One further voluntary option is available to those who complete this training with the same university. It is a two-year undergraduate course that is predominantly online, leading to a Bachelor of Arts in Applied Policing and Criminal Justice [78]. While the report acknowledges that the current recruit training program, which remains unchanged at the time of this paper, is further advanced than some other policing organisations, it does fall short of those of other democratic jurisdictions [30,79]. There, the process of the professionalising of policing is evidenced in a clear shift from 'training' to 'education', blended with a move from police institutional delivery to that incorporated directly within universities and Higher Education Institutions (HEIs) [30] (p. 70).

In response, Recommendation 30 of the TFPI report recommends a new learning and training strategy, which should incorporate recruitment, in-service training and CPD, supported by a ringfenced budget to deliver a developed educational and professional culture, all overseen by an expert review group inclusive of higher-level educational partners [30] (p. 70,), [80] (p. 10). The authors recommended that recruit induction must shift from 'training to education', in a move from university-validated courses delivered in house by police to 'direct involvement of [HEIs] in developing and delivering recruit training' in partnership, a reflection it observes of the professionalisation of policing internationally [30] (pp. 69, 70, 73). It was also recommended that police recruits who already hold degrees should spend less time in police training and should, rather, receive 'top-up' academic modules specific to policing, delivered directly by individual universities—a strategy that would also deliver cost savings according to the report. Universities would also be encouraged to both develop and directly deliver a policing

studies degree, that when successfully completed, would provide the swiftest route to becoming a sworn Garda. Those recruits without a degree would have the opportunity to attain a degree and become a sworn Garda, attending both the Garda Training College and universities [30] (pp. 69–73). While current recruitment is running at 800 Gardaí per year [30] (p. 67), to give additional focus to the possible future demand on HEIs, recently released figures indicate that between 2017 and August 2019, some 2090 Gardaí officers have been attested, while it is anticipated that a further 1500 will require attestation by 2021 [81], taking the force to its target figure of 15,000 officers by that date [82].

The TFPI report also advances the notion of 'in-service training' to that of in-service 'education' as a mandatory and accountable organisational requirement, designed to enable officers and employees to effectively carry out their policing roles and governed by the learning and training strategy. Differentiating from in-service training, it highlights that no overall CPD strategy existed and recommends its development, ensuring that all members have annually reviewed personal plans. To strengthen the overarching learning and training strategy, it directed a productive and robust review of partnerships with HEIs towards supporting all education and learning [80] (p. 10), [30] (p. 74,76).

Transitioning police training to higher education is a widespread phenomenon inextricably linked to reforms in the professionalisation of democratic policing [17,83–85]. Nonetheless, there is ambiguity in the relationship and the perceived benefits to policing, including that highlighted in Cox & Kirby's recent study challenging the proposition that for prospective police officers who engage in higher education broader, benefits will inevitably follow. Whilst citing Wood [86] (p. 276), who suggests that '[r]ecruits have an exposure to the external influence of an open campus . . . thereby limiting potential entrenchment of the negative culture which might be encouraged if their entire training was conducted in a closed and isolated residential college', they argue that specific policing degrees within a university setting can strengthen the relationship among the students, who are more likely to identify with a police culture and distance themselves from other students [5] (pp. 14–15). More widely, an argument exists that, as increasing numbers of officers professionalise through higher education, the police will fail to adequately reflect society [5,87].

Paterson [88] suggests three conceptual pillars from Marenin's [6] (p. 109) prerequisites for democratic policing, that of 'professionalism, legitimacy and accountability', as adapted yardsticks of the 'added value' that higher education delivers to police education and training. It is argued that such added value will only be achievable when the design and implementation of a learning strategy is supported by a robust evidential base tied to defined learning objectives [88] (p. 19). Heslop [4] cautions this hybrid police education, through the lens of Bourdieu's [1] related concepts of field, capital, habitus and symbolic violence, as evidenced in a United Kingdom (UK) study of police recruits engaged in a collaborative police and university foundation degree. He contends that universities hold the 'status' and academics the 'capital', this manifesting itself with as much 'repelling' as 'attraction' [4] (p. 9), highlighting the criticality of what he terms the development of the 'professional habitus'. He also identifies complexity within such hybrid approaches, in that student officers could find themselves in universities having had no formalised education, as entry to the police may not be based on academic qualifications [4].

Similarly, Wood & Tong report the tension that can exist between HEIs and police interests over the issue as to 'who owns' the student officers [89], and for the students themselves, an unclear identity emerges (between that of a police officer and a student), creating further conflicts of interest [5] (p. 7). In contrast, Huey's [90] systematic review (2000–2015) of in-service police training had to be halted due to inadequate numbers of peer-reviewed, published research papers on the worth of police training models on any one topic. Relative to this, Brown [91], when updating and extending Paterson's [88] narrative literature review of the value brought to policing by graduate officers as opposed to non-graduate officers including the value of specific policing or criminal justice degrees, fell short of finding conclusive evidence that higher levels of education delivered improved policing outcomes, which she viewed more as an indictment on the lack of research available [91]. All this suggests that there is unease between the police and universities as stakeholders in the education

of student police officers, with students often attaching to an ambiguous and conflicting identity (as neither student nor police officer), all set against inconclusive evidence that the higher educational habitus will typically deliver advantageous professional policing attributes.

However, if Heslop's ideal 'professional habitus' does indeed exist [4], one might look no further than the findings of Canterbury Christ Church University (CCCU) [92] in its case study of professional police education and training (PET) for a framework:

- The professionalisation of policing requires a validated, verified core of knowledge responding to changing environments.
- This core of knowledge must be derived via a 'critical friendship' with the world of academia.
- The assessment and accreditation of PET is essential, including, critically, the use of problem-solving models and a requirement to emphasise professionalism and ethics.
- The pedagogic requirements for the delivery of PET are made complex when the two worlds of higher education and police training converge.
- Finally, there is an emphasis on the development of national consistency in the linkage between police and higher education.

The CCCU case study evidences the 'impact' that these critical factors have had on communities, individuals and organisations that play crucial roles in PET, including the strategic direction of UK and European discussions around the professionalisation of policing through education and its delivery [92] (pp. 3–4). Moreover, in highlighting the 'significance' of their findings, the authors have added weight to debates in the UK, including contributing to the Independent Commission on Policing [93], concerning initial police training and education [92].

The reform and processes recommended by TFPI are now presented within a four-year implementation plan published in December 2018 and referred to as 'A Police Service for the Future' (PSF). Delivery of the plan will be governed by the newly established 'Implementation Group on Policing Reform' (IGPR), which will have two working links to the Department of the Taoiseach (Prime Minister). The first of these is to the 'High-level Steering Board on Policing Reform' (HSBPR), designed to assist with obstructions experienced in implementing the plan, and, secondly, to the newly formed 'Policing Reform Implementation Programme Office' (PRIPO). It is the responsibility of the latter to report to the HSBPR quarterly on the process of implementation and annually publish public reports on progress. Unfortunately, no annual progress report has been published, which does hamper research; however, PSF indicates that the first year is given over to the 'Building Blocks' and 'Launching' phases [94] (pp. 5–6).

### 2.4. Police Professionalism

Schinkle and Noordegraaf [95] (pp. 68, 71), reviewing Bourdieu's [1] (pp. 241–242) vigorous denunciation of the concept of 'profession', note that his rejection may appear startling when set against the mass of sociological literature of the early twentieth century on professions, leaving observers to question if the Bourdieusian approach to professionalism is feasible at all. However, they argue that it is conceivable to uphold Bourdieu's contention against such weight, if professionalism is viewed as a form of symbolic capital in what Bourdieu [1] (pp. 55–56) terms the 'field of power', stressing that adopting Bourdieu's approach facilitates a feasible and empirically useful conceptualisation of professionalism, as Bourdieu in his rejection of 'professionalism' facilitates an explanation of how power is established, appropriated and exploited in a manner that strengthens interpretation [95] (p. 71). While Brenhm et al. [96] describe professionalism as complex and a challenge to define, Clarke observes 'professional' as a 'slippery and overused term', contesting that there are two crucial characteristics of what it is to be a professional, firstly, that of expertise, including specialised knowledge, learning and skills, underpinned by ethical behaviours [3]. The second, critically flowing from the first, involves the esteem in which the public perceives the profession to be held, which within the context of this paper

could arguably be viewed as an essential element to those public perceptions of police legitimacy, perhaps suggesting a relationship between that legitimacy and Clarke's views on professionalism.

Studies of police officers' understanding of professionalism reveal their commitment to a high service ideal as well as temperate support to self-regulate, along with higher levels of professionalism demonstrated by entry-level officers [97]. Current literature also identifies education and training as key to police professionalisation and modernisation [98] (p. 8), [84] (p. 281), evidenced in the spread of police partnerships with academia and universities throughout the USA, Europe and the UK, developing training, education, research and knowledge transference [99]. Within Ireland, the TFPI is categorical that policing should not be seen as a job but as a profession supported by a professional culture and essential traits, including that of a cultural commitment to CPD [30] (p. 85). It identifies the greater involvement of HEIs in the education processes as a means for enhancing professionalism [30] (pp. 85–86), reflecting Neyroud's [100] recommendations on police professionalism in the UK, which focused on partnerships between the police and HEIs. While Neyroud [100] defines the police as a profession not unlike that of medicine or law, as they share sets of traits, such as a CoE, Holdaway [101] (p. 17) observes it as a benign approach to measuring a profession using a list of traits and argues that it overlooks the broader social context of the developed entitlements of the profession. In this context, claims to professionalism are neither expressed or examined nor are the meanings of these claims taken into account. Lumsden [98] also proposes that the trait-based approach to professions is discredited by Abbott's [102] broader sociological account of the systems of professions, avoiding issues raised by the traits-based approach, while assisting in understanding the complex and multiple views of the police themselves. Abbott points to the link between a profession and its work as a jurisdiction, which is contested by other occupations and professions in rivalry, raising the core question: who has 'control of what, when and how' [102] (p. 9)?

While Lumsden [98] identifies a growing body of literature on the professionalisation of policing, especially in the USA, Europe and Australia [103–105], Holdaway contends that academics tend to accept the police as a profession. He argues there is an uncritical and under-investigated acceptance by academics, who rely on an implied belief that the police 'might be and probably are a profession' [101] (pp. 1, 4–5), ignoring the broader social understandings of professions. Lumsden and Evetts [98,106] both declare that the professionalisation of the police can be a contested space, with tension being derived from top-down notions of what professionalism should be, the bottom-up thoughts and conceptualisations of the rank and file, and the external direction given by the state as to what professionalism in policing should resemble. Both Manning and Heslop take the view that police professionalism is employed by the police as a strategy to uphold their mandate and advance their authority, autonomy and self-worth [107,108] (p. 314). Thursfield [109] also links police professionalism to a need to influence society, echoing a belief that the public expect the police to act professionally [103]; therefore, in exhibiting professionalism, officers feel they can achieve recognition of their work by other professionals, the state and the public [110]. Vaidyanathan proposes that professionalisation requires a sense of belonging [111], while Lumsden [98] (p. 15) observes that the police identify themselves as professionals through perceived privilege and status linked to their recruitment, qualifications, and education, and if faced with a necessity to achieve this status within the organisation, they must show success in those areas, supported by the use of evidence-based knowledge (developed in conjunction with universities) and the exercising of new public management principles. Clarity in defining the term professional in a policing context does appear elusive, supporting Clarkes contention that 'professional' is indeed a slippery term [3].

The police can and do question the legitimacy of the organisational model and the political agenda that often underpins it, as it challenges their self-held belief as professionals. Some contest the managerial model and the external attempts to drive the new professionalisation agenda [98] (p. 17). In Ireland, this tension has most strikingly been demonstrated in recent months as 48% of the 12,000 Gardaí trained on the Code of Ethics refused to sign the code [112].

Holdaway's research suggests that police officers 'seem to assume that the police are already a profession and then go on to talk about professionalising the service' [101] (p. 3). Moreover, while some police officers may resist top-down reforms in policing, the marking of the police as professional is perceived as raising its esteem in the eyes of both the public and politicians and brings benefits for recruitment both in the terms of quality of applicants and their diversity [98] (p. 16), [113]. Lumsden observes in her research that professionalism can be viewed as a technology of control and discipline, propelling the occupational reform of police function and culture and suggesting that it is an amalgam of the following:

- The external political motivation to professionalise policing;
- Police education towards professionalism and recruitment towards diversity;
- An evidence base built from research between academia and police–academic partnerships towards the professionalisation of policing;
- The principles of new public management, including performance management and efficiencies;
- Ethics of policing [98] (pp. 10–14, 16).

It is to the last point on Lumsden's conception of professionalism, that we turn next, namely, the vital application of ethics in policing.

*2.5. Police Code of Ethics*

Neyroud et al. [114] (pp. 3–4), identify an array of external and internal problems as policing evolves and argue strongly for the merging of human rights with policing as a critical requirement for the continued progression and success of policing. They point to the comments of the Independent Commission on Policing for Northern Ireland [13], where they note the following:

> . . . a central proposition of this report that the fundamental purpose of policing should be . . . the protection and vindication of the human rights of all . . . policing means protecting human rights. [13] (p. 18)

Neyroud et al. place particular emphasises on the contention that 'ethical policing' with human rights at its core is vital to the reform process [114] (p. 4), while the earlier ICPNI report recommended a Code of Ethics as being critical to the integration of human rights into all police practice [13] (p. 20). England and Wales soon followed this lead, introducing a CoE in 2014, which is now theoretically rooted in the national decision making model, as the acid test to all police decision making [115,116], and is viewed as another step to full professional status for policing, comparable to that of medicine and law [115] (p. v). However, concerns have been raised as to whether it genuinely has become embedded across the 43 English and Welsh forces [117] (pp. 6–7). It is important to note once again the importance of conceptions of the police as a 'profession' versus their perceived need to 'professionalise'. For example, Lumsden [98] (p. 14), in her research, notes that the CoE in England and Wales may not be viewed by police officers as a means to professionalise policing, as they already view policing as a profession, citing Abbott [102], that occupations can be successful in pursuing professional standing as they are proficient in defending claims to a specialised knowledge. Lumsden also identifies a reliance on this specialised knowledge in defending professionalism, against the 'top-down' technology of professionalism [98] (p. 14).

In Ireland, the TFPI [30] (pp. xv, 12, 85) introduced an amended CoE, ensuring that Garda move from 'respecting' to 'acting' in a manner consistent with human rights, again reflecting the embedding of human rights theory into police practice, supported by a significant shift in training [114,118]. Despite the TFPI recommendation that the organisation must embrace the revised Code of Ethics [30] (p. 12), it was confirmed in 2019 that 48% of the 12,000 Gardaí who have received training on the Code of Ethics had actively failed to sign the code [112]. The Garda Representative Association (GRA) blamed the reluctance to sign on perceived training deficiencies, which they argue hinders Garda in their ability to deliver on the code, with the association quoting the following language:

> Ethics come with professionalism and as such there is an onus on An Garda Síochána
> as a responsible employer to provide adequate, continuous and up to date professional
> development, training...In the absence of organisational support to achieve professional
> competency, the Code places an unfair burden on the individual member. [119]

The Garda Commissioner has now refused to promote Gardaí who have failed to sign the Code of Ethics [120], raising questions around the culture within AGS to the point where AGS Policing Authority are considering an inquiry [121]—all evidence which may suggest that an existing occupational culture is a key barrier to reform.

### 2.6. Police Occupational Culture

> Cultures are the complex ensembles of values, attitudes, symbols, rules, recipes, and practices,
> emerging as people react to the exigencies and situations they confront, interpreted through
> the cognitive frames and orientations they carry with them from prior experiences. [8]
> (p 116)

While Bowden [21] (p. 3) refers to Reiner's [8] definition of a 'cop culture', he expands further, citing policing literature illustrating the affinity of the police to involve in 'group think' [21] (p. 3), which he defines as a distinctive occupational police culture, which is designed to shield the organisation from perceived external threats, such as examination by distrusting citizens, which, in turn, causes them to become ever more resistant to public scrutiny. The history of AGS, he concludes, reflects the extent to which it has been shaped by its own culture [21]. Charman and Corcoran observe that this culture is not something that AGS 'has', but, rather, it characterises informally what the organisation 'is', contesting that the reform of Irish policing is challenged by this culture and that it is simply not the straightforward implementation of improvements to policing [25] (p. 489). Connolly also contends that AGS was occupied by a 'cop culture', suspicious of outsiders and fused internally when facing external challenges, which was obstructive in the investigation of wrongdoing [72] and a barrier to accountability [6]. The independent 2018 report 'Play Your Part, Cultural Audit of AGS' delivered similar findings as the Commission itself principally identified an organisation distant from §being a professional service [30] (pp. 83, 84) [122].

Gundhus [104] contends that new regimes often face resistance, not just from the stubbornness of police culture, but also from the perceived threat to existing professional systems. In reality, police culture does not conform to homogenous depictions [123,124], varying within and between forces, including the social and political context in which they operate [125,126]. Police training institutions are fundamental to the socialisation of officers; therefore, a key lever to transforming police culture is to reform the culture of such institutions [53] (p. 23), [5] (p. 3). Goffman [127] observes the re-socialisation of police recruits, contending that it occurs in 'total institutions', described as places where individuals' physical and social freedoms are restricted. It is defined as a two-step process: first, the 'mortification of self', during which in individual attitudes, views and behaviours are stripped away, and second, a 're-socialisation', during which new values and beliefs are provided [127]. This mirrors the recruitment and initial training process of police training facilities, reflecting 'total institutions', where new officers interact with more experienced officers, reinforcing a new set of values, which brings them into line with the ethos and cultures of the institution [128] and which is reflected in the following words of Conlon:

> The day the new recruit walks through the door of the police academy, he leaves society to
> enter a profession that does more than give him a job, it defines who he is. [129] (p. 9)

The TFPI report identifies a similar 'closed culture' of separation from those outside the force, which has its geneses within recruitment training in AGS [130], the majority of which is delivered by other sworn officers. It recommends that further curriculum development and immersed delivery with HEIs will create an open and professional police culture [30] (p. 70). Specifically, when exploring

the prospects for cultural transformation within AGS, Mulcahy attaches importance to police training content in challenging the inappropriate loyalty of AGS officers both to themselves and the organisation over that of their responsibility to citizens [24] (p. 5).

## 3. Discussion and Conclusions

While AGS has traditionally enjoyed the widespread support of communities over the last decade, the research would suggest that its public legitimacy has actually been in contest, given the various policing controversies that have only served to undermine both public confidence and the legitimacy of policing [21,24,72]. This contested legitimacy was compounded internally and externally by changing perspectives in the delivery of 21st-century democratic policing [30] (p. 102), [24]. In this context, the CFPI are firmly focused on the development and delivery of education for AGS, central being the learning and training strategy; however, it is beset with challenges, including those emanating from the culture of policing, the varied characteristics of police recruits and even cultural and political factors from within HEIs [4]. Cox & Kerby reinforce this view, observing that the culture of policing is so intense that it can have a negative effect on reform, service delivery and police education. To counteract this within higher education programs, they see the need to strengthen student resistance to police culture both in the pre-recruitment stage and throughout a policing career [5]. They also argued that further consideration should be given to the minority of individual student officers within their research who contested and resisted the negative attitudes of police culture [5] (pp. 12–13). The strength and benefits of such an educational strategy, while not being totally discounted, must be placed in the context of inconclusive research outcomes from the recent studies of Brown [91], in relation to the value of graduate officers and the 15 year systematic review by Huey of police training, which failed to reach a conclusion due to inadequate research in any one area [90]. These findings point more towards an inadequacy of the research and critical examination, as opposed to the value of higher education [5] (pp. 2, 12).

It is important to give recognition to a significant driver for such an educational strategy—that of professionalism. Neyroud [100] sees education and, specifically, the forming of partnerships with HEIs as key to the development of professional policing. Therefore, it is telling that evidence of these intended HEI partnerships can be observed across AGS reforms. In terms of the UK experience, Brown's [91] (pp. 9–11) aforementioned highlighting of a lack of available empirical evidence ensures that we cannot draw a definitive conclusion as to the value of HEI partnerships to UK policing. However, the opportunity for conducting empirical research in the unfolding police educational reforms in the ROI certainly has the potential to deliver valuable evidence for England and Wales in their own progression towards 2020, when all new officers will have to be educated to the degree level [131–133], and for the subject of police education more broadly, as it develops importance for police organisations across the world [5].

According to Cox & Kerby [5] (p. 3), citing Worden and Butler & Cochrane [134,135], police officers are said to be sceptical, authoritarian, hardened, conservative, loyal, secretive, suspicious and isolated, presenting unethical behaviours, eroding citizens' confidence in policing and acting as a barrier to organisational reform [136]. In this context, the cultural audit of AGS delivered some not unexpected results [122] and is supported by the 2019 Monitoring Report of the Policing Authority, which found little evidence of a positive cultural change in AGS [137] (p. 8). The AGS cultural audit, in quoting Peter Drucker that 'culture eats strategy for breakfast' [122] (p. 50), raises concerns about the implementation of the key AGS learning and training strategy and shares the doubts previously raised by Cox & Kerby as to whether higher education has the capabilities to dilute police culture [5] (p. 4). If cultural positives exist, then it may be in the contention of Charman and Corocoran [25], who observed within their research that the influence of the occupational culture of AGS on individual officers responding to reform presented as much less insidious, harmful and compelling than might have been typically suggested in studies of police culture. They argue that the Gardaí should not be seen as 'cultural dopes' [138] (p. 500) but, rather, as a group who displays an ability to select what is best

from a 'cultural tool kit' [138] (p. 500), concluding that their research moves the debate beyond that offered by Manning—that AGS is 'an organisation very resilient and resistant to change' [18] (p. 347). However, further cautionary tones are raised by Conway, who identifies that little empirical research has been carried out on the police culture of AGS, although that which has been done indicates that it is 'at play'. She calls for further research and confrontation of the internal culture of AGS supported by strong leadership [34]. While the potential threat presented by police culture is clearly recognised, it does emphasise the need both in AGS and other policing jurisdictions for the requirement of any police educational strategy to be strong in design and flexibility to respond to the influence of police culture.

While AGS is focused on the role of Gardá being seen as a profession and not a 'job', Clarke's model in adaption and suggestion potentially presents an analytical lens with which to examine professional learning for the police, sensing its relationship to the perceived public legitimacy of policing while critically not ignoring the wider social context of the police professional, which both Lumsden and Holdaway identify as enriching that understanding [98] (p. 5), [101] (pp. 4, 5, 17, 27). As the TFPI reaffirms, 'From the time of recruitment, police education should not only teach the duties and responsibilities of police officers … It should also instil in members of the police service the cultural values we expect to see in our police—high ethical and professional standards' [30] (p. 69). However, it is impossible to ignore the evidence that police occupational culture presents a significant impediment to such reform; however, as Paoline et al. [54] suggests, concentrating efforts around enabling officers to engage positively in community policing may hold more potential in reform as opposed to changing the attitudes of officers. They envisage a developed police environment including supervision, training in key skills and staff appraisals as a means for supporting and not inhibiting community policing. It is suggested that the successful incorporation of a 'policing with the community model' and ethos within an educational and professional framework would be the most effective means for resisting an erosive occupational culture and ultimately supporting improved police legitimacy. While as yet not tested in this context, Clarke's model is potentially structured and flexible enough to provide future analysis on a synthesis of such a process and the other variables highlighted within this paper.

What clearly presents itself from the research is the most far-reaching policing reform to take place in Irish history and the first democracy of the 20th century. Its overarching objective is to professionalise AGS, thereby improving and maintaining its failing legitimacy, which historically has been fraught and offset by the reoccurrence of scandals and corruption. It has been argued that the police culture of the Garda coupled with their close alliance to the state have left them impervious to previous reforms. Mulcahy has also argued that the institutions established in earlier reforms to monitor and provide oversight to policing, such as the GSOC, were weakly equipped for their purpose from the onset due to a lack of political will [24]. Despite the attempts of these earlier reforms to militate against police malpractice, further scandals did reoccur, leading to both policing and political crises. However, the TFPI and its processes of implementation have deliberately integrated the Department of the Taoiseach into the process, securing political tethering to the reform implementation and AGS. It is around this model of political binding of AGS, the implementation of the reform process and the politicians themselves where further research may reveal its effectiveness or otherwise and allow reflection more broadly on such processes for democratic police reform.

One of the principle reforming strategies set out to achieve the 'professionalisation' of AGS is the 'education' as opposed to 'training' of Garda officers both at the time of recruitment and through CPD as they progress in their careers. The TFPI has recommended a full and strategic working partnership with HEIs in the design, input and delivery of education recruits and Garda officers. However, evidence posed on the education of police students in HEIs has been revealed as inconclusive, presenting opposing arguments regarding the experience and its value to the policing student, with one contention being that it can resist the pervasive and negative nature of police culture, while in contrast others

have argued that universities, in fact, can isolate police recruits further, strengthening and accelerating the establishment of police culture. However, consider Heslop's argument as follows:

> [T]hat professionalisation is not just about police officers having letters after their names, but much more critically it is about the need for the development of what might be termed 'professional habitus'. If universities are to be involved in education of police recruits, then this is also one of their most important roles. [4] (p. 12)

It would appear that the university will continue to be a vital cog in any such process with Heslop [4] prefacing his argument with the acknowledgement that he himself initially would have had little hesitation in agreeing with Lee and Punch on the importance of university education:

> Policing needs to be continually enriched with critical, enquiring and challenging minds. Uniformity and conformity lead to stereotypical thought and conduct that undermines this. A sound university education still provides the best basis for this thought. [139] (p. 248)

However, in summary, he not only identifies a policing culture but also that of a separate and distinct culture held by universities and contends that '*some*' academics are a part of that culture, and as such, it is not always positive. He explains that his research moved to employing a Bourdieusian lens to examine his results using the related concepts of field, capital, habitus and symbolic violence, which he argued provided an '*explanation in broader and relational terms*' [4] (p. 12). It could therefore be debated that adopting this Bourdieusian platform could be a critical element to researching the successful design of a university program more suitable for police recruits. Secondly, researching and developing such a program with elements of the additional thinking, arguments and findings identified, could provide a refined focus of thinking towards designing a more suitable '*professional habitus*'.

Some of the issues described above can be viewed in the work of Cox & Kirby [5], who engaged with 84 students on a two-year foundation degree in policing (FDP), available to police students only in a UK University, which itself delivers a range of criminal justice-related degrees to a total of 383 students. Their main findings included the following:

- The 84 police students were identified as being different and behaving differently to the other students, moving to social isolation and loyalty to their own cohort.
- The decision to enrol in an FDP programme clearly altered the way in which they viewed their own identity and the manner in which they behaved.
- The emergence of a confusing self-identity was found—neither student nor police officer—generating conflicts of interest and leading them to distance themselves from sensitive or problematic situations or lifestyles.
- They increasingly viewed themselves as police officers and socialised more with their own cohort, withdrawing from other student associations.
- Lecturers on the FDP were either police officers or ex-police officers with academic experience. Students disclosed that they were clearly influenced by these lecturers, with credibility stemming not from academic achievement but more from their stories and experiences as police officers. Critically, Cox & Kirby raise the question, can this be combatted by using 'pure academics' devoid of police experience?
- The students felt they spent more time at lectures and studying than other students.
- Their lecturers differentiated from their operational police trainers, who were more assertive and cut corners.
- As the course progressed, students displayed a changing attitude towards certain sections of the public, developing a 'them and us' attitude.
- Changing attitudes developed on human rights and diversity, with many observing that the police as an organisation was too politically correct.

- Students themselves began to recognise that a policing culture existed and varied between police teams and areas, and for some, this was a barrier to their own acceptance.
- Recognition by police students that policing was a lifestyle choice presented moral and ethical choices, forcing them to isolate themselves from other university students.

While Cox & Kirby acknowledge that immersion in university education does arguably deliver benefits to police students that could serve them well as they go on to engage with communities, building policing legitimacy, their study did highlight the shifting manner in which the university experience caused them to view themselves and how others perceived them.

They argue that it is simply not a question of moving police education from police training institutions to a university campus to change policing culture. The broader thinking, therefore, could be an acceptance that police culture exists in either habitus; nevertheless, universities potentially should present more convincing outcomes. It is therefore vital to design university programmes that recognise both the police and university cultures but in an innovative and creative manner that can compensate for such impediments and best conceive that 'professional habitus'. Paterson's argument that the value of higher education for the police rests within the design and implementation of an evidenced-based learning strategy that meets clearly defined goals [88] (p. 19) adds further support to the contention that if higher education programmes are to pursue this 'professional habitus', they should be evidence-based and subject to rigorous academic research.

As described earlier in this paper, further work should also be undertaken to consider how higher education can be extended further to fill shortfalls in the current level of training for serving officers. In addition to the 700 sworn AGS detectives who were assessed as being critically undertrained [32,43], recent evidence obtained by the Garda Professional Standards Unit (GPSU) in 2018 established that, although 81% of Gardá interviewed were involved in the taking of DNA samples, only 55% had received some form of training on the taking, submission and retention of samples [140]. It was also reported critically that some of the training was received via a 'how-to' video via a Garda portal, while representatives from the GRA states that it is only recruit officers from 2015 onwards that have been adequately trained [141,142], stating the following:

> We are screaming for this training for years. The Garda Representative Association puts high importance on training, it should not be a luxury, the report that has come out in relation to Garda training for DNA testing is not a surprise. [141]

Such findings are likely to be evident across other important areas of policing practice, pointing to a potentially strong need for additional training of Garda. Certainly, the range of potential measures discussed in this paper serves to highlight an opportunity to further develop these officers through the delivery of specialised modules within a university setting, supporting that movement of ethos from 'training to education' and 'job to profession'.

Finally, there is a clear need for empirical research on the reform process detailed in this paper, and perhaps, this is one good reason why police education needs to take place within a university context, (assuming of course that this is the best arena in which to facilitate evidence-based learning). Moreover, perhaps the police, including AGS, as in other professions, should be trained as researchers and research their own profession. Within a RoI context, this could be conceptualised in an independent National Police Research Institute, which could blend together experienced police and academics (both local and international) in a clearly defined role. Their aim would be to research what is best for the education and development of the police professional at this critical time of live and significant police reform in a democratic society. In providing research, guidance and support locally to both universities and AGS, it would have the opportunity to add significantly to the important knowledge base on the education and professionalisation of the police.

Overall, the recent unfolding of the reform of AGS in Ireland presents an opportunity to research and examine the transformational process further to improve practice and promote learning going forward.

**Author Contributions:** J.T. contributed to the design, writing, re-writing and editing of this research. B.P. contributed to the design, writing, re-writing and editing of this research.

**Funding:** This research has received no external funding.

**Conflicts of Interest:** The authors declare no conflicts of interest.

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
