# Peer review of "Towards Professionalism and Police Legitimacy? An Examination of the Education and Training Reforms of the Police in the Republic of Ireland"

_education, doi:10.3390/educsci9030241_

Round 1

Reviewer 1 Report

This is a fairly interesting paper but somewhat niche paper – I am not yet convinced of its utility beyond RoI. To improve, it would be helpful to see the paper begin with a fuller introduction, detailing the background to the research and, for readers who are less familiar with RoI, the Garda Siochana. The conclusion should spell out the implications of these findings for other police services.

Author Response

Response to Reviewer 1 Comments

Point 1:This is a fairly interesting paper but somewhat niche paper – I am not yet convinced of its utility beyond RoI. To improve, it would be helpful to see the paper begin with a fuller introduction, detailing the background to the research and, for readers who are less familiar with RoI, the Garda Siochana.’

Response 1: The following has been referred to in the manuscript in response:

Additional research detailing the background of the Garda Siochana (AGS) emphasising its close association to the establishment of the RoI. Its failing policing legitimacy, critical worldwide to all democratic policing, is highlighted along with its resistance to previous reforms. The current professionalisation of policing in the democratic world through education within university environments is identified as reflected within the professional and educational reforms of the AGS. Further evidence which suggests the benefits of such university education to police students and policing is identified as being inconclusive, possibly due to the lack of empirical research. More broadly, tensions associated with police culture and the education of police students in universities within other democratic jurisdictions is presented as relevant. The potential value of further research within the RoI as educational and professional reforms roll out in partnership with universities is highlighted as a fresh opportunity to add to the broader debate around police education and professionalisation.

Point 2:The conclusion should spell out the implications of these findings for other police services.’

Response 2: The manuscript has been revised to include and highlight the potential implications for other policing organisations, including:

Evidence of the continuing shift to educate police in a university setting. Evidence for the need for CPD, to be delivered as an ‘educational’ necessity for a professional policing service. The necessity for strong educational strategies (evidenced based), partnered with Universities and HEI’s, pursuing the professional development of police and the support of police legitimacy. Highlighting the pervasive and resistant nature of the police culture within both police and university environments. The argument that universities themselves have a particular culture that plays its part in the process of police educational reform. Additional research into the relationship between police students and universities may yield broader explanation when examined through a Bourdieusian lens. Consideration that the implementation of policing reform can potentially be weakened by an early absence of the political will to initially empower the mechanisms of reform as fit-for-purpose. Continuance of the argument that an evidence-based approach is required for the development of police education. The thread that education in enhancing the competency of officers to effectively deliver a community-based policing approach within their communities supports critical police legitimacy. That police themselves should be trained as researchers and work alongside academics developing police educational reform and professionalisation. Consideration of the evidence-based development of a ‘professional habitus’ which militates the potentially corrosive impact that police and university culture may have. Adaption and adoption of Clarke’s ‘Place Model’ as an analytical tool to support further research into the professionalisation of policing through education and its relationship to police legitimacy.

Reviewer 2 Report

The paper “Towards professionalism and police legitimacy? An examination of the education and training reforms of the Police in the Republic of Ireland” describes the process of reforms in the Ireland police well and in detail, so that reader can get a good insight into why are these reforms needed and how are they being implemented.

Highlights are well-written. The introduction contains all essential elements in order to "invite" the reader to further read and engage with the text. Literature review predominately involves all-important studies on the topic. But the last chapter Discussion and Conclusion is in comparison to other parts of the paper short and without exact practical implications, and it is missing some critical opinions about the reform. To be even more specific, some quantitative data should be added to how many police personnel will the reform impact, what is the need of Ireland police for the new personnel and how they should be train-educated (These numbers are the foundation for Educational programs). You have highlighted the importance of Higher education programs on the topic of police education that need to be introduced into the education system in the Republic of Ireland. From your sentence, in conclusion, it looks like there is no Higher education program in Ireland to improve the education, but there is at least one program  two-year, undergraduate, level 8 degree in Applied Policing & Criminal Justice from the University of Limerick (https://www.ul.ie/cpe/node/1831) which represent a continuation in training-education from An  Garda  Síochána  Training  College. So, your conclusion should be more specific. From the numbers in staff requires a preposition of new University programs could be made and also how to extra educate already active police personnel as you nicely stated that there is ‘’ 700 untrained detectives’’ who could be enrolled into these programs or attend specialized modules created for them which would count as specialized training-education.‘’ Also, an important factor should be addressed, if a reform like this should be successful, which is qualified Higher Education Teaching Personnel for those programs. Creating new University programs just to say we have them and ‘’to thick the reform box’’ without highly qualified Lecturers is not going to improve Professionalism nor Legitimacy of police personnel in the country. So, the focus should also be to identify highly knowledgeable individuals from within the police that could be further educated (abroad on similar programs) and could then help in this University programs (or to help the programs with visiting lecturers from similar Police systems). As you mentioned, research is also essential, and you could clearly state that the programs should include research methods as it is crucial for the police personnel who finish the University programs to be educated in this field. As they are then able to carry out much needed researching this field. Maybe to put out the idea of introducing a National Police Research Institute that could study the most urgent topics? Reforms are a long-distance run, not a sprint, so more clear suggestions should be made to help steer them in the right direction. The conclusion should also summarize the status of the reform (at what stage is it) and if there is no data also highlight that.  

Round 2

Reviewer 2 Report

Dear Authors,

I have read your revisited paper, and I will recommend to Editor to accept it.

Kind regards